# Intracranial-Pressure-Monitoring-Assisted Management Associated with Favorable Outcomes in Moderate Traumatic Brain Injury Patients with a GCS of 9–11

**DOI:** 10.3390/jcm11226661

**Published:** 2022-11-10

**Authors:** Mingsheng Chen, Haiyang Wu, Zhihong Li, Shunnan Ge, Lanfu Zhao, Xingye Zhang, Yan Qu

**Affiliations:** Department of Neurosurgery, Tangdu Hospital, Air Force Medical University, Xi’an 710038, China

**Keywords:** moderate traumatic brain injury (mTBI), intracranial pressure (ICP), monitoring, neurological deterioration (ND), outcome

## Abstract

Objective: With a mortality rate of 10–30%, a moderate traumatic brain injury (mTBI) is one of the most variable traumas. The indications for intracranial pressure (ICP) monitoring in patients with mTBI and the effects of ICP on patients’ outcomes are uncertain. The purpose of this study was to examine the indications of ICP monitoring (ICPm) and its effects on the long-term functional outcomes of mTBI patients. Methods: Patients with Glasgow Coma Scale (GCS) scores of 9–11 at Tangdu hospital, between January 2015 and December 2021, were enrolled and treated in this retrospective cohort study. We assessed practice variations in ICP interventions using the therapy intensity level (TIL). Six-month mortality and a Glasgow Outcome Scale Extended (GOS-E) score were the main outcomes. The secondary outcome was neurological deterioration (ND) events. The indication and the estimated impact of ICPm on the functional outcome were investigated by using binary regression analyses. Results: Of the 350 patients, 145 underwent ICP monitoring-assisted management, and the other 205 patients received a standard control based on imaging or clinical examinations. A GCS ≤ 10 (OR 1.751 (95% CI 1.216–3.023), *p* = 0.003), midline shift (mm) ≥ 2.5 (OR 3.916 (95% CI 2.076–7.386) *p* < 0.001), and SDH (OR 1.772 (95% CI 1.065–2.949) *p* = 0.028) were predictors of ICP. Patients who had ICPm (14/145 (9.7%)) had a decreased 6-month mortality rate compared to those who were not monitored (40/205 (19.5%), *p* = 0.011). ICPm was linked to both improved neurological outcomes at 6 months (OR 0.815 (95% CI 0.712–0.933), *p* = 0.003) and a lower ND rate (2 = 11.375, *p* = 0.010). A higher mean ICP (17.32 ± 3.52, t = −6.047, *p* < 0.001) and a more significant number of ICP > 15 mmHg (27 (9–45.5), Z = −5.406, *p* < 0.001) or ICP > 20 mmHg (5 (0–23), Z = −4.635, *p* < 0.001) 72 h after injury were associated with unfavorable outcomes. The best unfavorable GOS-E cutoff value of different ICP characteristics showed that the mean ICP was >15.8 mmHg (AUC 0.698; 95% CI, 0.606–0.789, *p* < 0.001), the number of ICP > 15 mmHg was >25.5 (AUC 0.681; 95% CI, 0.587–0.774, *p* < 0.001), and the number of ICP > 20 mmHg was >6 (AUC 0.660; 95% CI, 0.561–0.759, *p* < 0.001). The total TIL score during the first 72 h post-injury in the non-ICP group (9 (8, 11)) was lower than that of the ICP group (13 (9, 17), Z = −8.388, *p* < 0.001), and was associated with unfavorable outcomes. Conclusion: ICPm-assisted management was associated with better clinical outcomes six months after discharge and lower incidences of ND for seven days post-injury. A mean ICP > 15.8 mmHg, the number of ICP > 15 mmHg > 25.5, or the number of ICP > 20 mmHg > 6 implicate an unfavorable long-term prognosis after 72 h of an mTBI.

## 1. Introduction

Traumatic brain injury (TBI) affects people all around the world, with an estimated 10 million cases of TBI, particularly in low- and middle-income countries, resulting in hospitalization or death among young people each year [1,2,3]. A Glasgow Coma Scale (GCS) score of 9 to 12 at admission is commonly used to indicate a moderate traumatic brain injury (mTBI) [4], accounting for between 4% and 28% of TBI hospital admission patients [5].

One of the most typical secondary prognoses in moderate and severe TBIs is intracranial hypertension. Extensive cohort studies have shown that intracranial hypertension is an independent risk factor for poor outcomes and death in a severe TBI [2,6,7]. Although several studies have shown that elevated intracranial pressure is common in mTBI patients and related to poor prognoses [6,7,8,9], there is still insufficient direct clinical evidence on whether increased intracranial pressure leads to poor prognoses in patients with a moderate brain injury.

Intracranial pressure (ICP) monitoring has been utilized in neurosurgical practice for more than 50 years, and it is extensively used globally, especially in monitoring the intracranial pressure of severe TBIs [10]. Numerous studies have supported the benefits of ICPm technology in severe TBIs, and ICPm is associated with decreased mortality and favorable outcomes [11,12]. ICPm is advised for managing severe TBI patients with abnormal CT scans and special cases with normal CT scans based on the latest Brain Trauma Foundation guidelines [13]. For moderate TBI cases, no guidelines for ICPm are offered; nevertheless, studies have shown that mTBI patients with neurological deterioration have a mortality rate of up to 40%, especially in patients with an mTBI with a GCS score of 9–11 [9,14]. The leading cause of ND is intracranial hypertension induced by cerebral edema or delayed hemorrhage. If changes in intracranial pressure can be monitored, ND may be effectively observed and warned about. ICPm is an effective method with which to monitor changes in intracranial pressure. Still, there is no clinical evidence that ICPm is used to monitor and predict ND in mTBIs, and whether intracranial-pressure-monitoring-assisted management can improve neurological outcomes in mTBI patients remains unclear.

The objectives of this study were to evaluate the relationship between ICPm and lengthy neurological outcomes as well as the incidence of ND in mTBI patients with a GCS of 9–11. The relationship between early ICP characteristics and prognoses was also explored.

## 2. Patients and Methods

### 2.1. Design of the Study and Participants

Retrospective data collection was performed on patients with an mTBI who were hospitalized in our hospital’s neurological intensive care unit (N-ICU) between January 2015 and December 2021. The following were the inclusion requirements: (1) a GCS score of 9–11 at admission, (2) admittance 24 h after the injury, (3) a CT scan was performed the day after admission, and (4) patients aged >18 and ≤75 years old. The following were the exclusion requirements: (1) a penetrating injury’s presence and any related spinal cord trauma, (2) pregnant and breastfeeding women, (3) admission > 24 h post-injury, (4) patients who had evacuated mass lesions (as indicated by Marshall’s brain CT categorization) or who underwent craniotomies upon admission [15], (5) ICPm was not carried out later than 24 h after injury, and (6) the patient passed away the next day. The guidelines for treating severe TBIs were applied to all patients [13]. Based on the neurosurgeon’s expertise and the preferences of the patients or their legal representatives, an ICP monitor was implanted. Invasive ICPm will be recommended for all mTBI patients who meet the following three criteria: (1) a GCS of 9–11, (2) the presence of midline shift, and (3) cerebral contusions, or (and) SDH/EDH without surgical intervention.

### 2.2. Clinical Management

ICP-monitored and non-ICP-monitored patient groups were separated. ICP measurements were performed by using an intraparenchymal probe (Codman; Johnson & Johnson, Raynham, MA, USA; CAMINO. MPM−1; Integra Co., San Diego, CA, USA) inserted in the frontal cortex. Each patient was treated according to a standardized protocol: The vital signs and neurological status of all patients were monitored every 30 min within 6 h, after admission to NICU, and then every hour after 6 h. An imaging scan was performed 6 and 24 h after injury, and an imaging review was performed at any time if necessary. The treatment of elevated ICP episodes (ICP 20 mmHg for 10 min) followed a methodical approach to management [3]. The goal for cerebral perfusion pressure (CPP) was 60 mmHg. The non-ICP-monitored group’s ICP management was managed by the doctor’s experience based on imaging and clinical examinations.

### 2.3. Data Collection

Age, sex, primary diseases (hypertension, diabetes, chronic obstructive pulmonary disorder (COPD), and coronary heart disease), the background of medication (aspirin, clopidogrel, and anticoagulants), admission GCS score, mechanism of injury, and other demographic and clinical characteristics were recorded. Radiographic characteristics, such as epidural hemorrhage (EDH), midline shift, traumatic subarachnoid hemorrhage (tSAH), subdural hemorrhage (SDH), intraventricular hemorrhage (IVH), location of brain contusion (LOC), and so on, were recorded at admission. The modified Fisher (mFisher) scale was used to assess the severity of tSAH [16]. The criteria established by Marshall, L.F. were used to implement Marshall’s brain CT categorization [15].

In the hospital, the following variables were noted: ICP (estimated hourly on post-admitted days 1–5), therapy intensity level (TIL, ranging from 1 (minimum level) to 38 (maximum level)) [17] (measured daily on post-admitted days 1–5), adverse events, causes of neurological deterioration (ND), length of stay in the ICU and hospital, and 6-month neurological outcomes.

### 2.4. Outcome

The Extended Glasgow Outcome Scale (GOS-E) score [18], which evaluates the impact of mTBIs on patients’ function six months after discharge as well as the proportion of participants with unfavorable outcomes (GOS-E ≤ 4) and favorable outcomes (GOS-E > 4), served as the primary outcome measure. The secondary outcome was ND events. One or more of the following events occurred within seven days of the injury to qualify as ND: (a) a fall in GCS score of at least two points without the use of a pharmaceutical sedative from the baseline GCS score, and (b) a deterioration in a neurological condition that required neurosurgical intervention [9,19].

### 2.5. Ethics Statement

The study was approved by the Institutional Review Board of Tang Du Hospital, Fourth Military University (approval number: K202202–03), in February 2022. According to local and national legislation, the research ethics committee can be exempt from obtaining informed consent under specific circumstances.

### 2.6. Statistical Analysis

Using the Kolmogorov–Smirnov test, the data distributions of the two groups (ICP monitored vs. not monitored) were evaluated. The chi-square or Fisher’s exact test was used to compare baseline and ICP characteristics, and the *t*-test or Mann–Whitney U test was used to evaluate variables that were continuous. At a significance threshold of *p* < 0.05, a multivariate logistic regression analysis was carried out after the univariate analyses. The distributions of the 6-month GOS-E score distributions were compared using multivariable binary logistic regression. The resulting odds ratios (ORs) and 95% confidence intervals (CIs) were reported. A receiver operating characteristic (ROC) curve analysis was used to calculate the cutoff value and area under the curve (AUC) of different ICP characteristics. SPSS for Windows version 26.0 was used to conduct all statistical analyses (SPSS, Inc., Chicago, IL, USA). A two-sided *p*-value of α = 0.05 was used as the statistical significance level.

## 3. Results

### 3.1. Patients’ Demographic and Clinical Characteristics

A total of 350 patients participated in this study from January 2015 to December 2021. An overview of the patient selection process is illustrated in Figure 1. Of all of the patients, the median GCS score at admission was 10 (9–11); 145 (41.4%) patients had ICPm with GCS scores at admission of 10 (9–10); and 205 (58.6%) of the total patients did not undergo ICPm with GCS scores at admission of 10 (10–11). The median age of the study population was 54 years (42–63), and 255 of the population (72.9%) were male.

There were five (3.5%) patients who suffered from an ICP-related hemorrhage in the ICP group. The intracranial infection rate between the ICP and the non-ICP groups showed no significant difference (1.9% vs. 3.5%). However, patients who received ICP had a statistically significant shorter mean LOS in the ICU (4 (2, 9) vs. 6 (3, 10) days, *p* = 0.002) and in the hospital (8 (5, 13) vs. 10 (6, 12) days, *p* = 0.001) (Appendix A).

The GCS scores were lower in the patients who had ICPm than in those who were not monitored (Z = −3.593, *p* = 0.001). The patients who accepted ICPm showed a more remarkable midline shift (Z = −4.471, *p* = 0.001) and a higher prevalence of another location injury (ISS) than the group of non-monitored patients (Z = −3.718, *p* = 0.001). The patients who underwent ICPm had higher Marshall CT scales (χ^2^ = 11.653, *p* = 0.018) and modified SAH grades (χ^2^ = 12.915, *p* = 0.005). Additionally, the incidence of skull fracture and SDH in patients with ICPm was higher than in the non-monitored group. Appendix A shows the demographics and clinical characteristics of the patients included in the study. An ROC analysis was used to dichotomize continuous variables associated with outcomes, GCS, ISS, and midline shift (Appendix A and Appendix A, showing the detailed results).

Multivariate logistic regression showed that a GCS ≤ 10 (OR 1.751 (95% CI 1.216–3.023), *p* = 0.003), midline shift (mm) ≥ 2.5 (OR 3.916 (95% CI 2.076–7.386) *p* < 0.001), and SDH (OR 1.772 (95% CI 1.065–2.949) *p* = 0.028) were predictors of ICP (Appendix A).

### 3.2. Clinical Outcomes

Of the 350 patients, 54 (15.4%) suffered from death 6 months after discharge, of whom 14 (9.7%) accepted ICP monitoring and 40 (19.5%) were in the non-ICPm group. This difference was statistically significant. There were 229 (65.4%) patients who had favorable outcomes (GOS-E > 4) and 121 (34.6%) who had unfavorable outcomes (GOS-E ≤ 4). Among the 145 (41.4%) patients in the ICPm group, 108 (74.5%) patients had favorable outcomes and 37 (25.5%) patients had unfavorable outcomes; however, among the 205 (58.6%) patients in the non-ICPm group, 120 (58.8%) patients had favorable outcomes and 84 (41.2%) patients had unfavorable outcomes, and the difference in unfavorable outcome rates is statistically significant (χ^2^ = 9.176, *p* = 0.002) (Appendix A).

Univariate differential and inferential statistical analyses were used to find risk factors for unfavorable outcomes. Our study found that GCS scores, midline shift, ICP monitoring, Marshall’s scale, SDH, and location of contusion were correlated with unfavorable neurological outcomes (Appendix A). After adjusting for confounding factors, such as sex, alcohol abuse, ISS, midline shift (mm), tSAH-modified Fisher scale, and skull fracture, multivariate logistic regression showed that the effects of GCS scores (OR 0.463 (95% CI 0.312–0.686), *p* < 0.001), ICP monitoring (OR 0.815 (95% CI 0.712–0.933), *p* = 0.003), SDH (OR 2.115 (95% CI 1.243–3.599), *p* = 0.021), Marshall’s scale (type III DI, type IV DI, and NEML), and location of contusion (frontal, temporal, and frontal as well as temporal) on unfavorable outcomes were statistically significant (Table 1 and Figure 2). We also conducted a stratified analysis based on a GCS of 9–10 and a GCS of 11. In the GCS of 11 patient subgroup, among the 87 (34.4%) patients in the ICPm group, 60 (69.0%) patients had favorable outcomes and 27 (31.0%) patients had unfavorable outcomes; however, among the 166 (65.6%) patients in the non-ICPm group, 114 (68.7%) patients had favorable outcomes and 52 (31.3%) patients had unfavorable outcomes, and this was not statistically significant (Appendix A). In the GCS of 9–10 patient subgroup, among the 58 (59.8%) patients in the ICPm group, 42 (72.4%) patients had favorable outcomes and 16 (27.6%) patients had unfavorable outcomes; however, among the 39 (40.2%) patients in the non-ICPm group, 13 (33.3%) patients had favorable outcomes and 26 (66.7%) patients had unfavorable outcomes, and the difference is statistically significant (χ^2^ = 11.994, *p* = 0.001) (Appendix A). Additionally, multivariate logistic regression showed that ICP monitoring was an independent risk factor (OR 0.565 (95% CI 0.314–0.837), *p* = 0.001) (Appendix A).

Of the 350 GCS 9–11 mTBI patients, 131 (37.2%) developed ND within the first 7 days after admission. Among the 145 patients in the ICPm group, 39 (26.9%) had ND, and in the 205 patients in the non-ICPm group, 92 patients (44.6%) had ND within 7 days after injury, with statistically significant differences (χ^2^ = 11.375, *p* = 0.001) (Appendix A). Meanwhile, of the 131 ND mTBI patients, a GOS-E assessment of the patients’ prognoses showed 83 (63.8%) patients with a poor prognosis (GOS-E ≤ 4) and 48 (36.2%) patients with a good prognosis (GOS-E > 4). However, of the 219 non-ND mTBI patients, there were only 38 (17.4%) patients with a poor prognosis (GOS-E ≤ 4) and 181 (82.6%) patients with a good prognosis (GOS-E > 4). The statistical significance of differences between the two groups was significant (χ^2^ = 57.857, *p* < 0.001) (Appendix A).

Of all of the 350 patients, 46 (13.4%) had hematoma expansion, 83 (23.7%) had obvious cerebral edema aggravation, and 11 (3.2%) had other reasons for ND. Finally, of the 131 patients with ND, this was caused by hematoma expansion for 22 (52.4%) patients, by cerebral edema aggravation for 61 (78.2%) patients, and by general deterioration for 7 (63.6%) patients (causes of ND other than the above two reasons) in the non-ICP group. Still, it was caused by hematoma expansion for 20 (47.6%) patients, by cerebral edema aggravation for 17 (21.8%) patients, and by general deterioration for 4 (36.4%) patients in the ICP group; this difference was statistically significant (χ^2^ = 9.185, *p* = 0.010) (Appendix A). 

The total TIL score during the first 72 h post-injury in the non-ICP group (9 (8, 11)) was lower than in the ICP group (13 (9, 17), Z = −8.388, *p* < 0.001). Furthermore, the total TIL score during the first 72 h post-injury in the GOS-E ≤ 4 group (10 (9,13)) was lower than in the GOS-E > 4 group (12 (9, 16), Z = −3.000, *p* = 0.003), and in the ND group (10.5 (9, 13)) it was lower than in the non-ND group (12 (9, 16), Z = −2.571, *p* = 0.010) (Appendix A).

### 3.3. The Relationship between ICP Characteristics and Neurological Outcomes

Among the 145 ICPm patients, the minimum ICPm time was 24 h, the maximum time was 120 h, and the mean time was 71.0 ± 3.4 h. We analyzed the relationship between ICP of 72 h and unfavorable outcomes in ICPm patients.

Our study demonstrated that the mean ICP per hour was higher in the unfavorable outcome group than in the good outcome group (Figure 3A). The mean ICP of unfavorable outcome patients (17.32 ± 3.52) was also higher than that of favorable outcome patients (13.62 ± 2.91). The difference was statistically significant (t = −6.047, *p* < 0.001, Figure 3B). Meanwhile, the numbers of ICP > 15 mmHg frequency and ICP > 20 mmHg frequency were different in these two groups. The numbers of ICP > 15 mmHg frequency and ICP > 20 mmHg frequency in the unfavorable GOS-E group (27 (9–45.5) and 5 (0–23)) were more significant than in the favorable outcome patients (1 (0–9) and 0 (0–1)), and there is a statistically significant difference (z = −5.406, *p* < 0.001 and z = −4.635, *p* < 0.001, Table 2 and Figure 3C,D). The proportion of ICP > 15 mmHg spikes and the proportion of ICP > 20 mmHg in the unfavorable GOS-E group (37.50 (12.50–63.19%) and 6.94 (0–31.94%)) were also more significant than in the favorable outcome patients (1.39 (0–12.50%) and 0 (0–1.39%)), and there is a statistically significant difference (z = −5.406, *p* < 0.001 and z = −4.635, *p* < 0.001, Table 2).

To facilitate clinical application, the AUC and a diagnostic cutoff value of differential ICP characteristics were calculated by an ROC curve. The best unfavorable GOS-E cutoff value of different ICP characteristics showed that the mean ICP was >15.8 (AUC 0.698; 95% CI, 0.606–0.789, *p* < 0.001), the number of ICP > 15 mmHg was >25.5 (AUC 0.681; 95% CI, 0.587–0.774, *p* < 0.001), and the number of ICP > 20 mmHg was >6 (AUC 0.660; 95% CI, 0.561–0.759, *p* < 0.001) (Appendix A and Appendix A).

## 4. Discussion

In this study, the impact of ICPm on the neurological prognoses of mTBI patients with a GCS of 9–11 was retrospectively examined. First, we found that adjuvant therapy under ICPm was associated with a significantly better clinical outcome six months after discharge and a lower incidence of ND for seven days post-injury. Second, higher intracranial pressure treatment intensity (TIL score) was associated with a better six-month GOS-E. Finally, it showed that a mean ICP > 15.8 mmHg, a number of ICP > 15 mmHg more than 25.5, or a number of ICP > 20 mmHg greater than 6 at 72 h post-injury were distinctly associated with an unfavorable neurological outcome 6 months after discharge.

The most recent Brain Trauma Foundation guidelines suggest that ICPm should be used to manage severe TBI [13]. Still, the indications, type of monitoring device to be used, and optimal duration of the monitoring for mTBI patients are not clearly defined. ICPm is usually advised in patients at risk of increased ICP based on clinical and imaging findings as part of protocol-driven care [20]. In the previous literature, the indications for invasive ICPm in mTBI patients were as follows: (1) following surgery to remove several cerebral contusions or an acute subdural hematoma, (2) a GCS of 9–11 with (temporal or bifrontal) brain contusions without surgical treatment, (3) diffuse injury type III, (4) general anesthesia for urgent non-cranial surgery when substantial mass lesions are being treated conservatively (epidural, subdural hematomas, and contusions), (5) concomitant serious chest trauma necessitating prone ventilation, extensive sedation, high PEEP levels, or recruitment techniques, and (6) prolonged traumatic shock [21,22]. Our results revealed that mTBI patients who suffered from one or more of lower GCS, higher ISS, higher Marshall’s scale, have midline shift, SDH, higher mFisher level, and multiple cerebral contusions should be accepted for ICPm. Additionally, our study found that a GCS ≤ 10, midline shift (mm) ≥ 2.5, and SDH were independent risk factors of suffering ICP monitoring. These findings are consistent with the previous literature. The reason for these findings may be that most doctors think that these patients are more likely to develop ND and have poor prognoses. Nevertheless, obtaining an accurate ICPm indication for mTBI patients is extremely difficult, and many studies are needed to explore and confirm it.

ICPm is widely considered the cornerstone of TBI care because of the traditional view that suggests that clinicians can intercede with interventions to improve outcomes in mitigating secondary injury and death. However, the effect of ICPm on outcomes in TBI is controversial. Some studies have argued that ICPm is not an independent protective variable in terms of mortality and neurological outcomes, and that it is independently associated with increased overall complications [8,23,24,25,26]. Many studies have demonstrated that ICPm could reduce TBI mortality [11,27,28]. However, some other studies identified the fact that ICPm could reduce sTBI mortality and be associated with better functional outcomes under specific circumstances [12,29,30]. Several reasons may explain the discrepancy in the results in the literature. Some scholars believe that the severity of primary brain injury partially counteracts the positive effect of ICPm. Some experts think that ICPm is only a tool with which to guide treatment, and that it cannot change the outcomes of TBI patients [31,32]. ICP management based on monitoring may affect outcomes. However, most studies did not quantify the intensity of intracranial pressure management or explore the relationship between it and outcomes.

Another published retrospective study explored the effect of ICP on a GCS of 9–13 mTBI patients’ prognoses, and the results demonstrated that ICPm could hardly improve the functional outcomes of mTBIs but may possibly reduce in-hospital mortality [33]. This study has a relatively small sample size and the results may not be representative, since a GCS of 11–13 patients make up a large portion of all samples (61.5%). Conventional wisdom has it that patients with more severe TBIs are more likely to benefit from ICPm. Our research showed that ICPm was linked to better results after 6 months and a lower incidence of ND for 7 days post-injury in a GCS of 9–11 mTBI patients, especially for a GCS of 9–10 mTBI patients. A higher ICP management intensity (TIL score) was associated with a better six-month GOS-E and a lower incidence of ND. As we all know, mTBI patients have a lower primary injury degree than those with sTBIs, but have a higher probability of deterioration, require neurocritical care, and experience poor outcomes [6,7]. The current study’s findings demonstrated that mTBI patients with ND had more negative outcomes than individuals without ND. ICPm may change the outcomes by reducing ND by improving ICP management intensity (TIL score). In particular, our study showed that ICPm mainly reduced the proportion of neurological deterioration caused by cerebral edema aggravation. The deeper reason behind this may be that ICPm was more likely to detect fluctuations in ICP in a timely manner than the non-monitoring group based on imaging and clinical neurological status, thus providing physicians with an earlier opportunity for and strong intensity of treatment.

The most recent BTF guidelines have increased the threshold for ICP treatment from 20 to 22 mmHg [8]. However, the concept of the threshold has been discussed mainly by several authors, suggesting that the definition of a numeric threshold of ICP does not consider the complete pathophysiological features of brain-injured patients and the lack of high-level clinical outcome evidence [34,35]. Unfortunately, there is no accepted threshold for ICP treatment in mTBI patients. The present study demonstrated that early ICP characteristics are associated with an unfavorable GOS-E. Our study found that the mean 72 h ICP of mTBI patients with a GCS of 9–11 was approximately between 10 and 20 mmHg. Although the threshold did not reach 22 mmHg, the results suggested that a higher mean ICP and a larger number of ICP > 15 mmHg and ICP > 20 mmHg were associated with unfavorable outcomes. This is consistent with the results reported by Güiza et al. They found that even ICP insults at lower levels between 15 and 20 mmHg, if sustained, could lead to worse outcomes [36]. The possible reason for this may be related to the pathophysiologic heterogeneity of mTBIs. Furthermore, Launey et al. found that ischemia is common early, detectable up to 10 days, possible without high ICP, and associated with a poor prognosis [37]. Treatment strategies with a fixed ICP threshold need to be viewed with caution in mTBIs. Management individualization and determinations of optimal CPP or patient-specific critical ICP is the future direction.

The current study contains several flaws and limitations. Firstly, due to the study’s retrospective nature, some data might be missed or misinterpreted. There was just one center in this study, with a relatively small sample size. Secondly, ICP data available within 72 h do not fully represent ICP characteristics. Although we maintained CPP > 60 mmHg during the treatment process, we did not obtain ICP waveform and CPP data, which is a defect. Thirdly, the lack of data related to disposition, such as detailed later treatment measures and later rehabilitation treatment data, is a limitation. Finally, a small number of patients who deteriorated after seven days were excluded from this study, even though there were few patients in the population who experienced a late deterioration.

## 5. Conclusions

We conducted a retrospective study that found that ICPm was linked to a more favorable clinical result six months after discharge and a lower incidence of ND for seven days post-injury in mTBI patients with a GCS of 9–11. Multiple persistent mild elevated intracranial pressure was associated with a poor prognosis. The results provided important information for the ICP management of mTBI patients in the acute phase. However, further prospective studies are required to confirm our results.

## Figures and Tables

**Figure 1 jcm-11-06661-f001:**
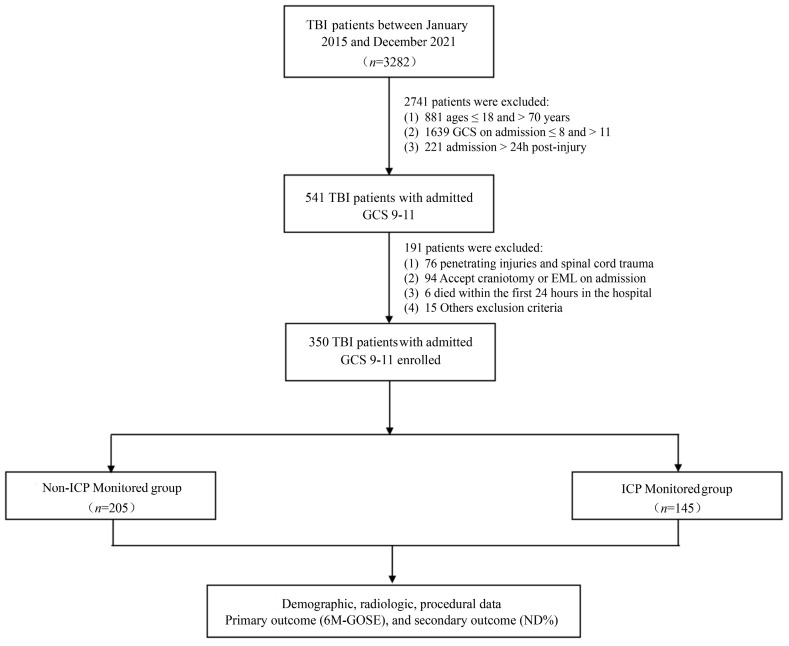
Flow chart illustrating the selection process for the study’s participants.

**Figure 2 jcm-11-06661-f002:**
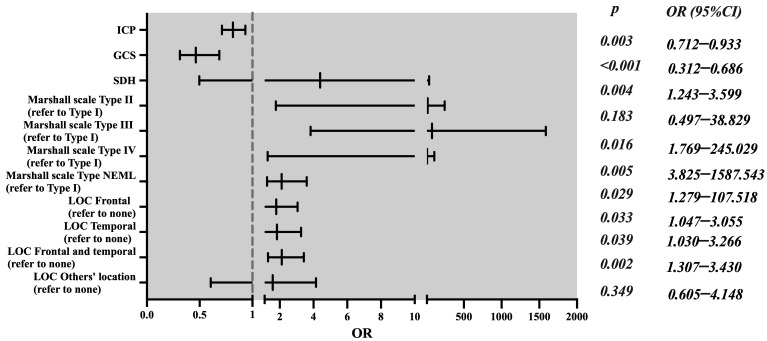
The forest plot of data from multivariate logistic regression, revealing factors associated with unfavorable outcomes (GOS-E ≤ 4) of mTBIs.

**Figure 3 jcm-11-06661-f003:**
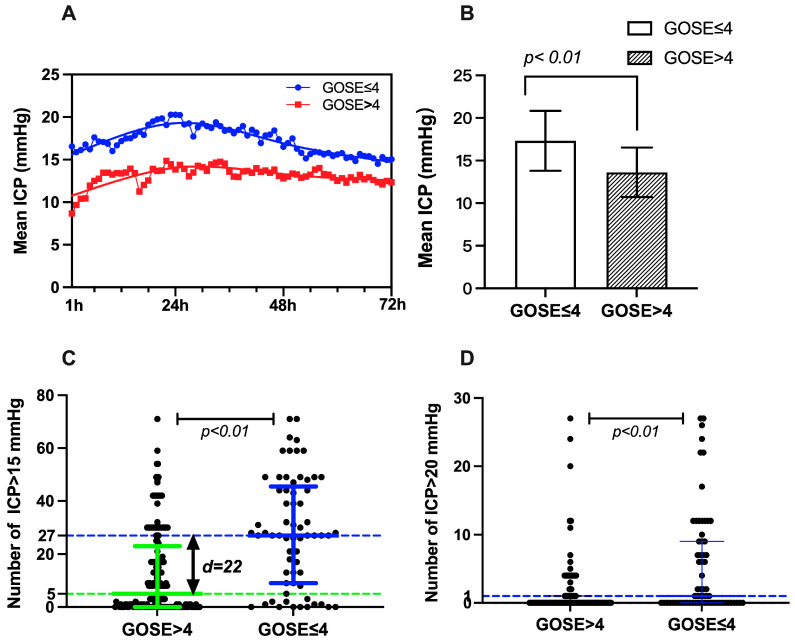
The relationship between ICP characteristics and GOS-E outcome. (**A**) The mean ICP per hour was higher in the unfavorable outcome group than in the good outcome group. (**B**) The mean ICP of unfavorable outcome patients (17.32 ± 3.52) was also higher than that of favorable outcome patients (13.62 ± 2.91) (t = −6.047, *p* < 0.001). (**C**,**D**) The numbers of ICP > 15 mmHg frequency and ICP > 20 mmHg frequency in the unfavorable GOS-E group (27 (9–45.5) and 5 (0–23)) were more significant than in the favorable outcome patients (1 (0–9) and 0 (0–1)) (z = −5.406, *p* < 0.001 and z = −4.635, *p* < 0.001) (The green line and the blue line show the median difference).

**Table 1 jcm-11-06661-t001:** The results of the multivariate logistic regression analysis of GOS-E ≤ 4.

Characteristics		OR	95% CI	*p*-Value
ICP monitoring	Yes	0.815	0.712–0.933	0.003
	No *	1		
GCS score		0.463	0.312–0.686	<0.001
Midline shift (mm)		1.144	0.982–1.323	0.057
Marshall’s scale	Type II DI	4.393	0.497–38.829	0.183
	Type III DI	20.821	1.769–245.029	0.016
	Type IV DI	77.928	3.825–1587.543	0.005
	NEML	11.725	1.279–107.518	0.029
	Type I DI *	1		
SDH	Yes	2.115	1.243–3.599	0.004
	No *	1		
Location of contusion (LOC)	Frontal	1.788	1.047–3.055	0.033
	Temporal	1.834	1.03–3.266	0.039
	Frontal and temporal	2.118	1.307–3.43	0.002
	Other location	1.584	0.605–4.148	0.349
	None *	1		

OR, odds ratio; 95% CI, 95% confidence interval; and *, control group.

**Table 2 jcm-11-06661-t002:** The differences of ICP characteristics between the GOS-E ≤ 4 and GOS-E > 4 groups.

	GOS-E ≤ 4	GOS-E > 4	Difference and 95% CI	Z/T	*p*
Mean ICP (mmHg) ± SE	17.32 ± 3.52	13.62 ± 2.91	3.70 (1.82–4.58)	−6.047	<0.001
Median number of ICP > 15 mmHg (IQR)	27 (9–45.5)	5 (0–23)	17 (10–23)	−5.406	<0.001
Median number of ICP > 20 mmHg (IQR)	1 (0–9)	0 (0–1)	1 (0–1)	−4.635	<0.001
Proportion of ICP > 15 mmHg % (IQR)	37.50 (12.50–63.19)	6.94 (0–31.94)	23.60 (13.90–31.90)	−5.406	<0.001
Proportion of ICP > 20 mmHg %	1.39 (0–12.50)	0 (0–1.39)	1.40 (0–1.40)	−4.635	<0.001

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
