# Peer review of "Intracranial-Pressure-Monitoring-Assisted Management Associated with Favorable Outcomes in Moderate Traumatic Brain Injury Patients with a GCS of 9–11"

_jcm, 2022, doi:10.3390/jcm11226661_

Round 1

Reviewer 1 Report

The manuscript “Intracranial pressure monitoring-assisted management associated with favorable outcomes in moderate traumatic brain injury patients with GCS 9-11” concluded that ICP monitoring was associated with better outcome and less frequency of neurological deterioration in moderate TBI patients. The results in this manuscript indicated that intracranial hypertension after moderate TBI is a risk factor related to poor prognosis. While more analysis or discussion about the data is necessary for the conclusion.

1.     As the author claimed, 145 patients with ICP monitoring and 205 patients without ICP monitoring were selected for this study. The implantation of the ICP monitor was based on the neurosurgeon’s experience and the choice of patients or their legal surrogates. Was there possibility that patients might refuse implantation of ICP monitor for financial reasons? therefor, they might not offer for the same medical treatment as others. The ICP monitored patients showed better outcome might because they got better and experienced medical treatments.

2.     From Line 205, authors showed that of the 131 patients with ND, 22/20(non-ICP/ICP) patients with hematoma expansion, 61/17 patients with cerebral edema, and 7/4 patients with general deterioration. For all the patient in this study, including patients without ND, are there differences between the non-ICP and ICP group about the incidence of cerebral edema, hematoma expansion?

Author Response

Reviewer #1:

Comment 1: As the author claimed, 145 patients with ICP monitoring and 205 patients without ICP monitoring were selected for this study. The implantation of the ICP monitor was based on the neurosurgeon’s experience and the choice of patients or their legal surrogates. Was there possibility that patients might refuse implantation of ICP monitor for financial reasons? therefor, they might not offer for the same medical treatment as others. The ICP monitored patients showed better outcome might because they got better and experienced medical treatments.

Response: It is true, as the reviewers suggest, that some patients will refuse to undergo ICP monitoring for financial or other reasons, but this is a very small minority. Each patient was treated according to a standardized protocol (the vital signs and neurological status of all patients were monitored every 30 minutes within 6 hours after admission to NICU, and then every hour after 6 hours. An imaging scan was performed 6 hours and 24 hours after injury, and an imaging review was performed at any time if necessary). The non-ICP-monitored group ICP management was managed by the doctor’s experience based on imaging and clinical examinations following a stepwise management strategy based on the guidelines for treating severe TBI (line125-133). In addition to the above, ICP management in the ICP monitoring group was combined with hourly ICP values to guide treatment. This only difference is the very factor we want to explore. Of course, as a retrospective study, it is difficult to achieve complete agreement on confounding factors, so we have conducted a prospective study of this project being enrolled patients (ClinicalTrials.gov Identifier: NCT04900168).

Comment 2: From Line 205, authors showed that of the 131 patients with ND, 22/20(non-ICP/ICP) patients with hematoma expansion, 61/17 patients with cerebral edema, and 7/4 patients with general deterioration. For all the patient in this study, including patients without ND, are there differences between the non-ICP and ICP group about the incidence of cerebral edema, hematoma expansion?

Response: Based on the reviewers' suggestions we did further analysis. Of all 350 patients, 46 (13.4%) had hematoma expansion, 83 (23.7%) had obvious cerebral edema aggravation, and 11 (3.2%) had other ND reasons. There are differences between the non-ICP and ICP groups regarding the incidence of cerebral edema, and hematoma expansion. For more information, see Supplementary Table 11.

Reviewer 2 Report

​I read this article with great interest. Moderate traumatic brain injury is a much-discussed topic in the literature and is certainly worthy of further investigation. The authors show that management supported by intracranial pressure monitoring is associated with an overall better outcome (both lower mortality and less neurological deterioration) in mTBI patients with GCS between 9 and 11. The article is well structured. An unconvincing aspect is the method used to separate the enrolled patients between monitoring with and without ICP. In particular, the choice of neurosurgeon may have led to a certain selection bias (lines 98-99). I therefore fully agree with the limitations of which the authors are aware. It could also be interesting if more space were given to the presentation of the results stratified by age, sex, and pathology of the included patients. There are some typos throughout the manuscript (e.g. the double space before the comma at the line 41)

Author Response

Reviewer #2:

Comment 1: I read this article with great interest. Moderate traumatic brain injury is a much-discussed topic in the literature and is certainly worthy of further investigation. The authors show that management supported by intracranial pressure monitoring is associated with an overall better outcome (both lower mortality and less neurological deterioration) in mTBI patients with GCS between 9 and 11. The article is well structured. An unconvincing aspect is the method used to separate the enrolled patients between monitoring with and without ICP. In particular, the choice of neurosurgeon may have led to a certain selection bias (lines 98-99). I therefore fully agree with the limitations of which the authors are aware. It could also be interesting if more space were given to the presentation of the results stratified by age, sex, and pathology of the included patients. There are some typos throughout the manuscript (e.g. the double space before the comma at the line 41)

Response: We agreed with the reviewer’s comments totally, reviewed the literature and found a large gap in prognosis between patients with GCS scores below 10 and those with scores above 10. We did a stratified analysis based on GCS scores and found that treatment under intracranial pressure monitoring was associated with good prognosis in patients with a score below 10, but not in patients with a score of 11 (Line 225-238) (Supplementary Table 6, 7, 8). In addition, we carefully checked the manuscript for typos and symbols and corrected them.

Reviewer 3 Report

On a series of patients with moderate traumatic brain injury, the authors compared a cohort with intracranial pressure (ICP) monitoring to a cohort without intracranial pressure monitoring. They found that 6-month outcome measured by Glasgow Outcome Scale-Extended to be better in the monitored group.

The manuscript suffers from several methodological flaws. Indications for ICP monitoring are not included, with the authors stating ICP monitoring was left to the discretion of the physician. This flaw is problematic, as most neurosurgeons and trauma surgeons would not advocate for ICP monitoring in this group. Why would monitors have been placed in the first place in so many patients? Possible indications are not even mentioned in the DISCUSSION. The group with monitoring has significant differences with the unmonitored group, including lower Glasgow Coma Scale and worse Marshall score. Site of hospitalization is not detailed; presumably the monitored patients were in an intensive care unit (ICU), but what about the unmonitored patients. No data on intervention for elevated ICP, and what that level might have been, is provided. No data related to complications of ICP monitoring were included. No data on mortality, ICU length of stay (LOS) or hospital LOS were included. No data on disposition (rehab versus skilled nursing facility, versus home) were included. The number of ICP elevations greater than 20 is not clearly reported, but the number appears to be a small proportion of the patients. These flaws beg the question that the results seen may just be an epiphenomenon of better adherence to treatment protocols in the ICP monitored group, an observation which has previously been made by others, but is not addressed in the DISCUSSION by the authors. In addition to addressing all of these flaws, the following more minor points should also be addressed:

1. In the ABSTRACT, in line 26, "variable treatments" is vague.

2. In INTRODUCTION, in line 59, moderate TBI is generally defined as 9-12, including in the reference cited.

3. In RESULTS, in lines 207 and 209, what is meant by "general deterioration." Could this deterioration have just been related to patient delirium, which has been associated with subsequent poorer outcomes in other disease processes?

4. In RESULTS, line 217 states that the minimum length of ICP monitoring was 20 hours. How is this time possible, when exclusion criteria in line 96 indicates patients with monitoring less than 24 hours were excluded?

5. In DISCUSSION, use of "enormous" in line 306 is overstated.

Author Response

Reviewer #3:

Comment 1: Indications for ICP monitoring are not included, with the authors stating ICP monitoring was left to the discretion of the physician. This flaw is problematic, as most neurosurgeons and trauma surgeons would not advocate for ICP monitoring in this group. Why would monitors have been placed in the first place in so many patients? Possible indications are not even mentioned in the DISCUSSION. The group with monitoring has significant differences with the unmonitored group, including lower Glasgow Coma Scale and worse Marshall score.

Response: We are very grateful for the comments we received from the reviewer. We fully agree with the reviewers' comments. We have added to the methods section our recommended indications for placing ICP monitoring (line118-120), and we also further analyze the risk factors for ICP monitoring in the results. We found that GCS≤10 (OR 1.751 (95% CI 1.216-3.023), P=0.003), Midline shift (mm) ≥2.5 (OR 3.916 (95% CI 2.076-7.386) P <0.001), SDH (OR 1.772 (95% CI 1.065-2.949) P=0.028) were predictors of ICP monitored (Line197-203) ((Supplementary Table 1, 2, 3 and Supplementary Figure 1).

Comment 2: Site of hospitalization is not detailed; presumably the monitored patients were in an intensive care unit (ICU), but what about the unmonitored patients.

Response: All patients were admitted to the NICU; we have explained this in the original manuscript. (Line106-108)

Comment 3: No data on intervention for elevated ICP, and what that level might have been, is provided.

Response: We give specific intracranial pressure management protocols in the methods section (Line125-133), and we recorded the therapy intensity level (TIL) score (Line145-146) to assess the intracranial pressure treatment intensity (reference 18). The results were on Line 259-264. (Supplementary Table 13, 14, 15)

Comment 4: No data related to complications of ICP monitoring were included. No data on mortality, ICU length of stay (LOS) or hospital LOS were included. No data on disposition (rehab versus skilled nursing facility, versus home) were included. The number of ICP elevations greater than 20 is not clearly reported, but the number appears to be a small proportion of the patients.

Response: We added data on the complications of intracranial pressure monitoring and the length of stay of patients and analyzed them. For detailed results, see Lines 184-188 and Supplementary Table 1. The data for this study were based on the inpatient medical record system. Rehab versus skilled nursing facility, versus home data collection was difficult. There is no doubt that this is an area we need to improve in the future. One of our authors, Zhihong Li, is writing an article to describe detailed clinical features of patients with elevated ICP over 20, which we did not explore further.

Comment 5: These flaws beg the question that the results seen may just be an epiphenomenon of better adherence to treatment protocols in the ICP monitored group, an observation which has previously been made by others, but is not addressed in the DISCUSSION by the authors.

Response: We added a similar observational study literature, analyzed it in the discussion, and cited it. (Line 316-319) (Reference 34: Li Z, Xu F, Li Y, Wang R, Zhang Z, Qu Y. Assessment of intracranial pressure monitoring in patients with moderate traumatic brain injury: A retrospective cohort study. Clin Neurol Neurosurg. 2020;189:105538.)

Comment 6: In addition to addressing all of these flaws, the following more minor points should also be addressed:

In the ABSTRACT, in line 26, "variable treatments" is vague.

In INTRODUCTION, in line 59, moderate TBI is generally defined as 9-12, including in the reference cited.

In RESULTS, in lines 207 and 209, what is meant by "general deterioration." Could this deterioration have just been related to patient delirium, which has been associated with subsequent poorer outcomes in other disease processes?

In RESULTS, line 217 states that the minimum length of ICP monitoring was 20 hours. How is this time possible, when exclusion criteria in line 96 indicates patients with monitoring less than 24 hours were excluded?

In DISCUSSION, use of "enormous" in line 306 is overstated.

Response:

“With a mortality rate of 10—30%, moderate traumatic brain injury (mTBI) is one of the most variable traumas.” This statement is intended to illustrate that mTBI injuries are prone to change and the prognosis is highly variable.

We have changed "9-13" to "9-12" (Line 73)

Indeed, the expression "general deterioration" is easily misunderstood. We defined “general deterioration” as causes of ND other than the above two reasons (hematoma expansion, cerebral edema aggravation) (Line 254). Because we find that some patients will have a reduced level of consciousness due to lung infections, post-traumatic stress, etc.

“the minimum length of ICP monitoring was 20 hours” was a writing mistake. We have corrected it as “24” (Line 266)

We have changed the word “enormous” to “large”. (Line 364)

Round 2

Reviewer 3 Report

The authors have revised their manuscript and have made improvements. A few additional changes would also be helpful to improve clarity, as follows:

1.  Should line 109 read "2) admittance within 24 hours of injury"?

2. In lines 118-120, were these recommendations followed for all patients? If not, lack of a strict protocol for ICP monitoring represents a limitation of the manuscript and should be so stated.

3. ICP monitoring should be differentiated from ICP. One suggestion is to abbreviate ICP monitoring as "ICPm". For example, line 184 is confusing in its current form.

4. Further discussion of the relationship of TIL to ICP monitoring or no ICP monitoring is worth mention in the DISCUSSION, as this data suggests that some patients with moderate TBI are under-treated.

5. Lack of data related to disposition is a limitation which should be discussed (not just in the context of imaging data as stated in line 379).

Author Response

Thank you for your letter and comments concerning our manuscript. Those comments are all valuable and very helpful for improving our paper.

Comment 1: Should line 109 read "2) admittance within 24 hours of injury"?

Response: There may be a misunderstanding here that is difficult to understand. Based on the inclusion and exclusion criteria of our study, we excluded patients who came to the hospital 24 h after the injury. Because we believe that early treatment is also critical, there is a possibility that it could be a confounding factor. To eliminate this misconception, we added an exclusion criterion to the exclusion criteria. (Line 109-112 and Figure 1)

Comment 2: In lines 118-120, were these recommendations followed for all patients? If not, lack of a strict protocol for ICP monitoring represents a limitation of the manuscript and should be so stated.

Response: These recommendations were followed for all patients if the patient meets the three criteria above. However, it is up to the patient and legal representative to decide. Whatever the patient and family decide, we will do everything we can to give the best treatment possible. We have corrected the original expression to avoid ambiguity. (Line115-117)

Comment 3: ICP monitoring should be differentiated from ICP. One suggestion is to abbreviate ICP monitoring as "ICPm". For example, line 184 is confusing in its current form.

Response: Thank you very much for your suggestion. We have changed the word "ICP monitoring" to ICPm in the full text.

Comment 4: Further discussion of the relationship of TIL to ICP monitoring or no ICP monitoring is worth mention in the DISCUSSION, as this data suggests that some patients with moderate TBI are under-treated.

Response: We have added relevant analysis and discussion, as detailed in Line339-350.

Comment 5: Lack of data related to disposition is a limitation which should be discussed (not just in the context of imaging data as stated in line 379).

Response: We fully agree with your comments. There is no doubt that this is an area we need to improve in the future. Of course, this is a limitation of this study. We have clearly stated such a limitation in the manuscript (Line374-375).

Finally, thank you again for giving us helpful suggestions and comments. This will guide us to be more standardized and accurate in our future studies.